# Effective implementation of the Sport Education Model in physical education: A meta-analysis of participant and intervention characteristics

Gege Yao[1], Junlong Zhang[2]*, Kim Geok Soh[2]*, Xiaorong Bai[3], Wensheng Xiao[3], Mohd Ashraff Mohd Anuar[4], Lixia Bao[2]

**1** The 19th Middle School, Qinhuangdao, Hebei, China, **2** Department of Sports Studies, Faculty of Education Studies, University Putra Malaysia, Seri Kembangan, Malaysia, **3** School of Physical Education, Huzhou University, Huzhou, China, **4** Department of Professional Development and Continuing Education, Faculty of Education Studies, University Putra Malaysia, Seri Kembangan, Malaysia

* zhangjunlong2413@gmail.com (JZ); kims@upm.edu.my (KGS)

## Abstract

Despite substantial evidence supporting the positive effects of the Sport Education Model (SEM) on students' physical abilities, mental health, and social skills, significant knowledge gaps persist regarding the moderating variables that influence its effectiveness. This study investigates the facilitative effects of SEM on students' physical education learning and examines the Participant and Intervention Characteristics that modulate its impact. Following the PICOS framework, two researchers independently conducted a comprehensive literature search across Web of Science, Scopus, PubMed, and EBSCOhost (CINAHL with Full Text and SPORTDiscus with Full Text) databases. The quality of the included studies was assessed using the Cochrane Handbook for Systematic Reviews of Interventions, and a meta-analysis was performed on the selected studies. A total of 15 studies involving 2,890 participants were included in this meta-analysis. The results indicate that SEM significantly improves students' physical education learning outcomes (Effect size = 0.590, 95% CI: 0.284–0.897, P < 0.001). Subgroup analyses revealed that SEM intervention was particularly effective for secondary school students (Effect size = 1.055, 95% CI: 0.361–1.759, P = 0.003), those in small class sizes (Effect size = 1.058, 95% CI: 0.314–1.802, P = 0.005), and students without prior SEM experience (Effect size = 0.604, 95% CI: 0.136–1.072, P = 0.011), and the most effective SEM intervention plan comprises 2 sessions per week (Effect size = 1.820, 95% CI: 0.486–3.154, P = 0.008), with each session lasting at least 60 minutes (Effect size = 1.002, 95% CI: 0.437–1.568, P = 0.001) and a total of no more than 18 sessions (Effect size = 0.654, 95% CI: 0.297–1.010, P = 0.001). SEM effectively enhances students' physical education learning and positively influences their cognitive and non-cognitive abilities. The most effective intervention includes two weekly lessons of at least 60 minutes each, totaling no more than 18 lessons, targeting secondary students without prior SEM

**Data availability statement:** All relevant data are within the paper and its Supporting Information files.

**Funding:** This study was supported by A Project Supported by the Education Ministry's Youth Fund Project for Humanities and Social Sciences Research of China (Grant No. 24YJC890001).

**Competing interests:** The authors have declared that no competing interests exist.

experience in small class sizes. This study offers practical recommendations for SEM implementation and theoretical support for the high-quality development of school sports.

## Introduction

The Sport Education Model (SEM) is a distinctive and well-structured pedagogical framework designed to provide students with authentic and developmentally appropriate experiences in physical education [1–3]. By focusing on enhancing students' abilities, confidence, and desire to engage in physical activity, SEM lays a robust foundation for lifelong active participation [1,4–6]. Unlike traditional physical education models that often emphasize skill acquisition in isolation, SEM adopts a student-centered approach, where learners take on a variety of roles—such as players, coaches, officials, scorekeepers, and equipment managers—within stable, homogeneous groups [7–10]. This multi-role involvement not only cultivates a comprehensive understanding of the sport, including rules, strategies, and tactics, but also fosters leadership, responsibility, and teamwork skills [5,11–13].

Guided initially by the teacher, SEM employs a progressive release of responsibility model [3,14]. The transition from teacher-directed instruction to more autonomous student-led practice empowers students to take charge of their learning [15]. This approach is realized through a series of engaging and challenging tasks that draw on students' strengths and interests, thereby promoting positive social dynamics, collaboration, and sustained motivation [7,16,17]. The model's structure includes five key stages: teacher-directed instruction, team selection, modified preseason activities, formal competition, and cumulative events. Each stage is designed to sequentially build both the cognitive and affective domains of learning [3]. Additionally, SEM's motivational environment is instrumental in supporting key components of physical literacy—including intrinsic motivation, self-efficacy, and enjoyment—thereby enabling students to acquire the confidence, competence, and movement-based literacy essential for lifelong physical activity engagement [18–20].

Despite substantial evidence supporting SEM's positive effects on students' physical abilities, mental health, and social skills, significant knowledge gaps remain concerning the moderating variables that influence its effectiveness [1,4,21]. Most existing reviews tend to generalize SEM's benefits without delving into how different factors might modify these outcomes [1,3,6,13]. For instance, it is still unclear whether students' responses to SEM vary significantly across different age groups (elementary, secondary, and university levels), or how factors such as students' prior experience with SEM and class size might influence their engagement and motivation. Moreover, the potential impacts of intervention frequency, session duration, and overall intervention period on learning outcomes are rarely quantified in current literature. Understanding these moderating effects is crucial, as they can substantially influence the effectiveness of SEM in promoting both cognitive and non-cognitive outcomes [3,6].

To address these gaps, this study conducts a meta-analysis to systematically synthesize existing data and examine the effects of SEM on student PE learning with a specific focus on two key moderating variables: Participant Characteristics (such as age, prior experience, and class size) and Intervention Characteristics (including frequency, duration, and period of interventions). By quantitatively analyzing these factors, this research aims to provide evidence-based insights into the optimal conditions for SEM implementation. Specifically, this study seeks to:

1. Determine how different participant characteristics influence the effectiveness of SEM in promoting cognitive and non-cognitive learning outcomes.

2. Determine how different intervention characteristics influence the effectiveness of SEM in promoting cognitive and non-cognitive learning outcomes.

3. Provide actionable recommendations for educators and policymakers on designing SEM-based interventions.

The findings of this study are expected to have significant implications for the field of physical education by offering a more granular understanding of SEM's effectiveness across different educational contexts. By bridging the current research gaps, this study not only aims to enhance the theoretical framework of SEM but also to inform policy and practice, enabling the design of more targeted and effective SEM interventions. In doing so, it contributes to the broader goal of high-quality development of school-based physical education, ensuring that SEM can play a transformative role in promoting lifelong physical activity and holistic development among students.

## Methods

This review follows the updated PRISMA guidelines [22] (See S1 Appendix. PRISMA checklist) and is registered on INPLASY (registration number INPLASY202530011). More information can be found at the following link: https://inplasy.com/.

### Search strategy

We conducted a comprehensive literature search across five electronic databases: Web of Science, Scopus, PubMed, and EBSCOhost (CINAHL with Full Text and SPORTDiscus with Full Text), with the search completed on February 6th, 2025. Customized search strings were designed for each database, using the following terms: ("Sport Education Model" OR "Sport Education" OR "Sport season") AND ("student*" OR "physical education" or "PE") AND ("performance" OR "outcome" OR "achievement" OR "effect*" OR "influence" OR "experiment" OR "academic achievement" OR "exercise capacity" OR "physical fitness" OR "physical quality" OR "knowledge" OR "skills" OR "motivation" OR "interest" OR "attitude" OR "mental health"). Additionally, a thorough manual search of Google Scholar and the reference lists of all included articles was conducted to ensure no relevant publications were overlooked. Detailed search strings for each database are provided in supplementary material S2 Appendix.

### Selection criteria

The inclusion criteria were established based on the PICOS (Population, Intervention, Comparison, Outcomes, and Study Design) framework [23], as follows: (a) there were no restrictions on the gender, age, or inclusion of vulnerable groups with physical disabilities, and there were no requirements regarding prior participation in SEM learning by the students; (b) the study must include a description of the interventions for both the experimental group (SEM) and the control group, with the interventions implemented in an educational setting and lasting more than two weeks. In addition, experimental interventions in which SEM is combined with other teaching models are allowed to be included, such as SEM integrated with Teaching Games for Understanding; (c) the intervention frequency, duration, and intensity for the control group must be similar to those of the experimental group; (d) the study must include at least one outcome measured using a valid, objective tool; (e) the research design must be a randomized controlled trial or a quasi-experimental study.

Exclusion criteria were as follows: (a) irrelevant populations, such as participants who are not currently enrolled students or those not involved in physical education courses; (b) not SEM intervention or if the hybrid pedagogical model was not explicitly based on the structural framework and elements of SEM, the study would be excluded; (c) studies with designs other than randomized controlled trials or quasi-experimental studies, and control groups that did not use standard or traditional teaching methods were excluded, such as SEM compared with other teaching modes, or mixed teaching compared with SEM; In addition, non-equivalent control studies will be excluded. (d) books, conference proceedings, theses, reviews, meta-analyses, retracted studies, erratum, and studies without available abstracts were also excluded (e) studies lacking sufficient data for analysis.

## Literature screening

Two researchers independently screened the literature and extracted data. Due to the large number of initially selected articles, only titles and abstracts were reviewed after duplicates were removed. Articles that did not meet the PICOS inclusion criteria were excluded. Subsequently, full-text readings were conducted for studies that potentially met the criteria, and studies that did not align with the research objectives and design were excluded. Finally, the two researchers (JLZ, GGY) cross-checked the screening results; in case of disagreements, a third researcher (WSX) was consulted.

## Risk of bias in individual studies and certainty of evidence

The RoB-2, the latest version of the Cochrane Risk of Bias tool, was used to evaluate the risk of bias for each randomized controlled trial included [24]. For quasi-experimental studies, the Risk of Bias in Non-randomized Studies of Interventions (ROBINS-I) tool was used to assess the risk of bias [25]. The certainty of the evidence was analyzed and summarized according to the guidelines outlined in the GRADE Manual [26]. Two research team members (JLZ, GGY) independently evaluated the risk of bias for each included study.

## Eigenvalue coding and data extraction

The characteristics of the included studies were systematically coded for subsequent analyses. In alignment with the focus of this research, the independent variable was coded as the Sport Education Model (SEM), while the dependent variable was coded as physical education learning efficiency. Modern developmental psychology indicates that the cognitive development of children and adolescents should be divided into different stages based on their growth processes. Students must progress through various cognitive development phases, ranging from concrete to abstract thinking and from perception and induction to reasoning [27]. Considering the patterns of student growth and development, an asynchronous relationship exists between their cognitive and non-cognitive abilities [28]. Therefore, the dependent variable was classified into cognitive factors (codified as A): academic performance, motor skills, physical fitness, theoretical knowledge, and practical skills; and non-cognitive factors (codified as B): learning motivation, learning interest, learning attitudes, and mental health [28]. Participant and intervention characteristics were also coded as moderator variables. Participant characteristics: number of students, learning stage, experience with SEM, and class size. Intervention characteristics were: the frequency of the intervention, the duration of the session, and the period of the intervention. The respective rules were: (1) school stage that was coded as the school of elementary, secondary, and university; (2) previous SEM experience coded as "yes", "no", and "not mentioned"; (3) class size coded as the small class (0–30 students), medium class (31–50 students), and large class (>50 students); (4) frequency of the intervention coded as ≤1 lesson/week, 2 lessons/week, and >2 lessons/week; (5) session length coded as greater than 60 min/session, 30–60 min/session, and less than or equal to 30 min/session; (6) the intervention period coded by one of the following: < 18 sessions and ≥18 sessions. In addition, research sources, the number of students, and intervention program were extracted.

### Statistical analyses

If three or more studies had sufficient data to compute the effect size (ES), we performed a meta-analysis [29,30]. Hedges'g (the effect size) is calculated as the means and standard deviation (SD) from pre- and post-measurement data of learning efficiency for the SEM group and control group. Data are adjusted for the post-intervention pooled standard deviation and a random-effects model is utilized so that underlying heterogeneity between studies could influence the SEM effect [29,31]. The ES values were accompanied by 95% confidence intervals (CI), and the ES values were interpreted according to the following scales: (a) values greater than 0.8 are considered "large," (b) scores between 0.5 and 0.8 are classified as "moderate," and (c) values between 0 and 0.5 are described as "small" [28]. For studies involving multiple different intervention groups, the sample size of the control group is divided proportionally to ensure that all group members are comparable [32]. If there is not enough data stated (e.g., missing in graphs or absent) we try to contact with corresponding author. If there is no response from the authors or they cannot provide the data, we exclude the study. The $I^2$ statistic is used to assess the heterogeneity of the studies, with values below 25% indicating low heterogeneity, 25%–75% representing moderate heterogeneity, and above 75% indicating high heterogeneity [33]. An extended Egger test is conducted to test the risk of publication bias [34]. The significance level used in the meta-analysis is $p < 0.05$ to establish statistical significance. All analyses are performed using Comprehensive Meta-Analysis software (Version 3.0; Biostat, Englewood, NJ, USA).

## Results

### Study selection

As shown in Fig 1, a total of 769 articles were retrieved from the databases, and an additional 4 articles were obtained through references and Google Scholar. After manually removing duplicate articles (343 articles), 430 articles remained. Following a preliminary assessment of the titles and abstracts of these records, strictly adhering to the inclusion and exclusion criteria, 60 articles were identified as eligible for full-text review. After a thorough evaluation of the full texts of these 60 articles, 45 were excluded, resulting in a final selection of 15 studies that met all the criteria for this meta-analysis (See S3 Appendix. Description of 430 records).

### Risk of bias in individual studies and certainty of evidence

Three cluster randomized controlled trials [20,35,36] were assessed using the ROBINS-2 tool, while 12 non-randomized controlled trials [7,8,21,37–45] were evaluated using the ROBINS-I tool. Among these trials, 2 showed an overall high (or serious) risk of bias [35,42], 11 demonstrated a moderate risk of bias [7,8,20,21,36,38–41,43,45], and only 2 exhibited a low risk of bias [37,44], as illustrated in Figs 2 and 3. The visualization of the ROB2 assessment results is presented in Fig 2. In the three cluster randomized controlled trials [20,35,36], none reported the methods for random sequence generation, merely stating adherence to randomization methods, resulting in a judgment of moderate risk of bias during the randomization process. One study [35] was identified as having a high risk of bias due to missing data, which was related to a dropout rate exceeding 10%. The visualization of the ROBINS-I assessment results is shown in Fig 3. One study [42] was deemed to have a high risk of bias due to re-grouping during the intervention process; the primary reasons for the moderate risk of bias in other studies were the lack of detailed reporting on missing participants or the absence of blinding of outcome assessors.

Table 1 presents the results of the GRADE analysis. The GRADE analysis revealed that the certainty of the evidence supporting the findings was very low.

### Study characteristics

The characteristics of the 15 included studies are presented in Table 2. Among them, 10 studies were published between 2020 and 2025, involving a total of 2,890 participants, with 1,318 in the experimental group and 1,572 in the control group.

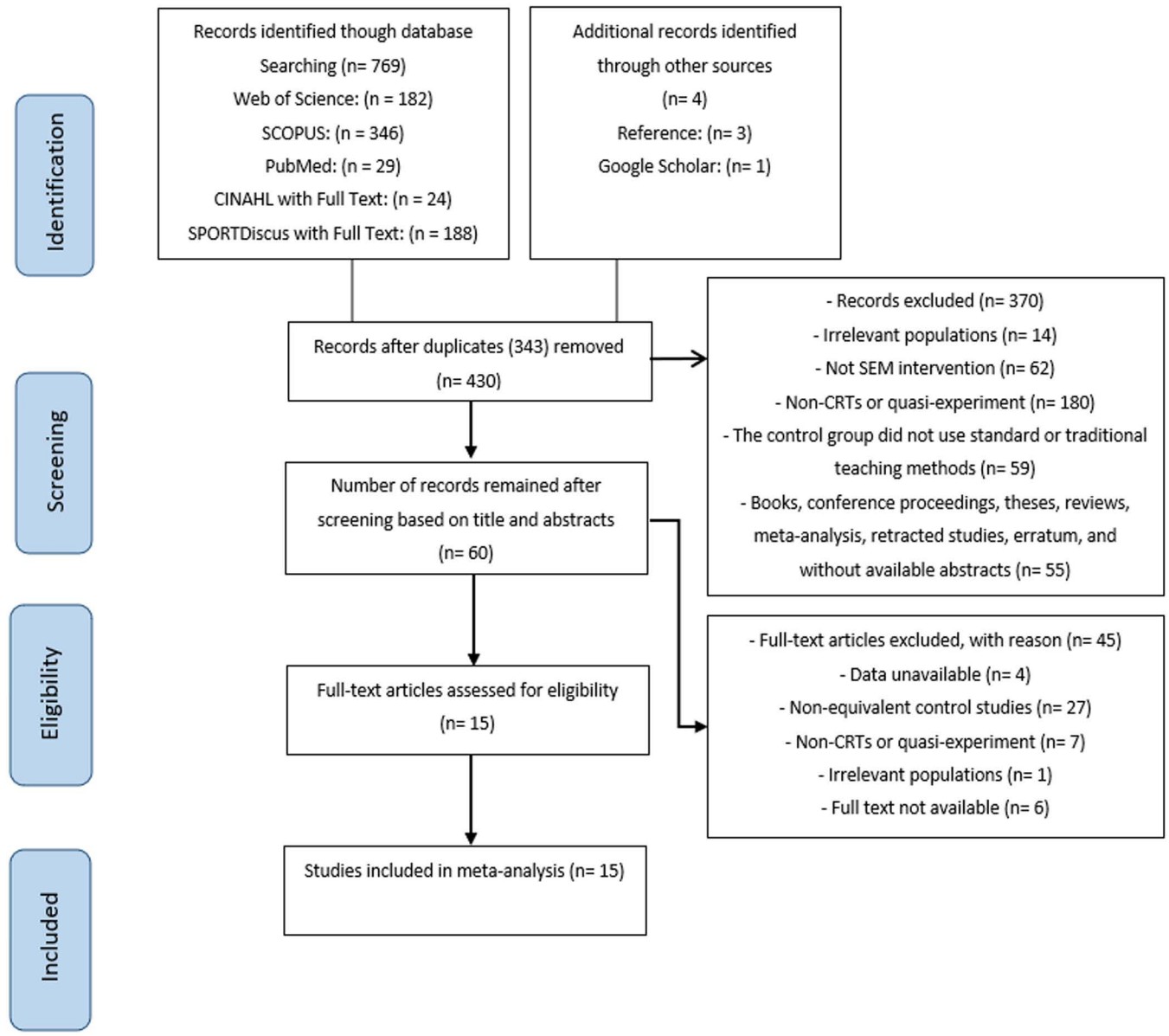

**Fig 1. PRISMA flow diagram.**

The participants consisted of 2 elementary-level studies [8,35], 9 secondary-level studies [7,21,36,38,39,41–44], and 4 university-level studies [20,37,40,45]. Regarding SEM experience, 1 study [42] included participants with SEM experience, 6 studies [8,36,37,39–41] included participants without SEM experience, and 8 studies [7,20,21,35,38,43–45] were classified as unable to determine due to a lack of relevant information in the articles. In terms of class size, 7 studies [36,37,39–41,43,44] were categorized as small, 4 studies [20,38,42,45] as medium, and 4 studies [7,8,21,35] as large. All experimental groups adopted the characteristics of the Sport Education Model (SEM) as the main framework, while the control groups employed standard traditional teaching methods. Regarding the intervention frequency, 4 studies [20,40,44,45] conducted ≤1 lesson per week, 6 studies [36–39,41,42] conducted 2 lessons per week, and 4 studies

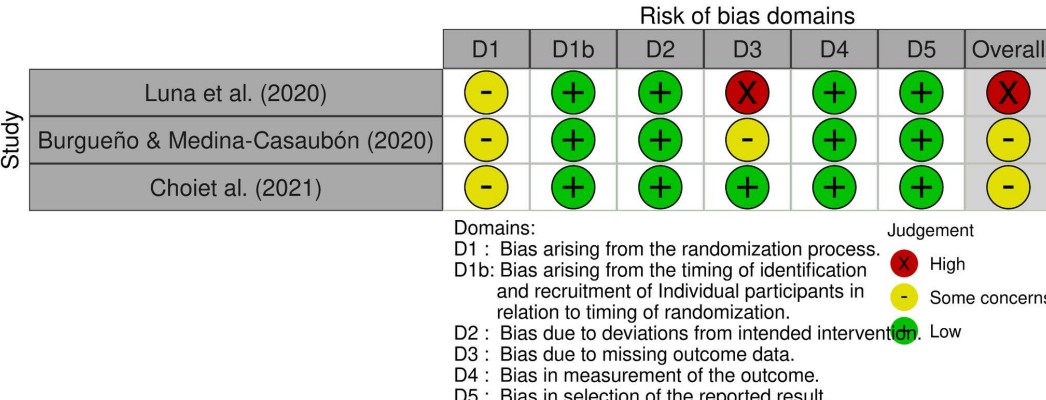

**Fig 2. RoB-2 (Cluster) assessments.** Created using the Robvis tool.

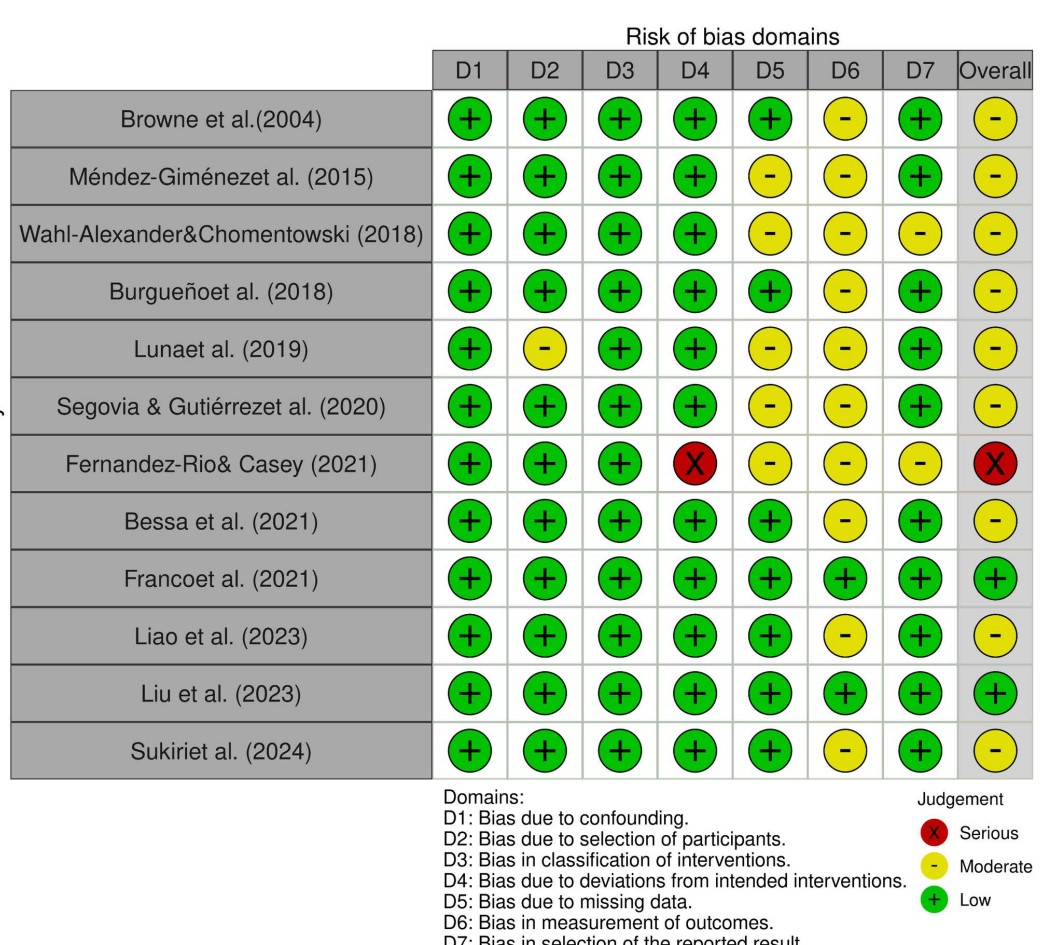

**Fig 3. ROBINS-I assessments.** Created using the Robvis tool.

**Table 1. GRADE analyses.**

| Outcomes | Certainty assessment | | | | | Number of participants and studies | Certainty of evidence (GRADE) |
|---|---|---|---|---|---|---|---|
| | Risk of bias | Inconsistency | Indirectness | Imprecision | Risk of publication bias | | |
| PE learning effect (cognitive dimension) | Serious[a] | Serious[b] | Not serious | Not serious | Serious[e] | 681 (5 studies) | ⊕○○○ Very low |
| PE learning effect (non-cognitive dimension) | Serious[a] | Serious[b] | Not serious | Not serious | Serious[e] | 2209 (11 studies) | ⊕○○○ Very low |

Note: High Certainty: Very confident the true effect is close to the estimate. Moderate Certainty: Moderately confident the true effect is likely close but could differ substantially. Low Certainty: Limited confidence; the true effect may be very different. Very Low Certainty: Very little confidence; the true effect is likely very different. GRADE Downgrading Reasons: a) Downgraded by 1 level due to high or some risk of bias. b) Downgraded by 1 level due to substantial heterogeneity (I2 ≥ 50%). c) Downgraded by 1 level for <400 participants or unclear effect direction. d) Downgraded by 2 levels for imprecision based on the above points. e) Downgraded by 1 level due to significant Egger's test (p < 0.05).

**Table 2. Study characteristics.**

| Study | Participant characteristics | | | | Intervention characteristics | | | | Learning efficiency |
|---|---|---|---|---|---|---|---|---|---|
| | Number of students (EG/CG) | Learning Stage | Experience with SEM | Class size | Intervention programme (EG/CG) | Intervention frequency | Duration of the session | Experimental period | |
| Browne et al. (2004) [39] | 27/26 | Secondary | No | Small | SEM/TT | 2 lessons/week | 30–60 min/session | ≥18 session | A |
| Méndez-Giménez et al. (2015) [7] | 185/110 | Secondary | UTD | Large | SEM/TT | UTD | 30–60 min/session | <18 session | B |
| Wahl-Alexander & Chomentowski (2018) [40] | 23/24 | University | No | Small | SEM/TT | ≤1 lessons/week | ≥60 min/session | <18 session | A |
| Burgueño et al. (2018) [41] | 22/22 | Secondary | No | Small | SEM/TT | 2 lessons/week | 30–60 min/session | <18 session | B |
| Luna et al. (2019) [21] | 69/44 | Secondary | UTD | Large | SEM/TT | >2 lessons/week | 30–60 min/session | <18 session | B |
| Luna et al. (2020) [35] | 85/57 | Elementary | UTD | Large | SEM/TT | >2 lessons/week | 30–60 min/session | ≥18 session | B |
| Segovia & Gutiérrez et al. (2020) [8] | 59/44 | Elementary | No | Large | SEM/TT | >2 lessons/week | 30–60 min/session | <18 session | A |
| Burgueño & Medina-Casaubón (2020) [36] | 74/74 | Secondary | No | Small | SEM/TT | 2 lessons/week | ≥60 min/session | <18 session | B |
| Choi et al. (2021) [20] | 188/184 | University | UTD | Medium | SEM/TT | ≤1 lesson/week | ≥60 min/session | <18 session | A & B |
| Fernandez-Rio & Casey (2021) [42] | 48/42 | Secondary | Yes | Medium | SEM/TT | 2 lessons/week | 30–60 min/session | <18 session | B |
| Bessa et al. (2021) [43] | 204/226 | Secondary | UTD | Small | SEM/TT | >2 lessons/week | ≥60 min/session | <18 session | B |
| Franco et al. (2021) [44] | 25/25 | Secondary | UTD | Small | SEM/TT | ≤1 lesson/week | 30–60 min/session | <18 session | B |
| Liao et al. (2023) [45] | 49/46 | University | UTD | Medium | SEM/TT | ≤1 lesson/week | ≥60 min/session | <18 session | B |
| Liu et al. (2023) [37] | 56/50 | University | No | Small | SEM/TT | 2 lessons/week | ≥60 min/session | ≥18 session | A |
| Sukiri et al. (2024) [38] | 204/226 | Secondary | UTD | Medium | SEM/TT | 2 lessons/week | ≥60 min/session | <18 session | B |

[8,21,35] conducted >2 lessons per week, with one study [7] not reporting the frequency of intervention. For the duration of each session, no study reported ≤30 minutes per session, 8 studies [7,8,21,35,39,41,42,44] reported 30–60 minutes per session, and 7 studies [20,36–38,40,43,45] reported ≥60 minutes per session. In terms of the experimental period, 12 studies [7,8,20,21,35,36,38,40–45] involved <18 sessions, while 3 studies [35,37,39] included ≥18 sessions. The learning outcomes assessed included cognitive indicators (such as sports ability and physical activity) in 6 studies [8,20,37,39,40,42] and non-cognitive indicators (such as motivation) in 10 studies [7,20,21,35,36,38,41,43–45].

### Synthesis of the results

**The overall implementation effect of the Sport Education Model in physical education teaching.** The assessment of the overall implementation effects of the SEM in physical education involved all 15 included studies, comprising 16 experimental groups and 15 control groups. Some studies measured multiple cognitive and non-cognitive indicators, totaling 49 (11 cognitive indicators and 38 non-cognitive indicators) (See S4 Appendix. Date used for meta-analysis for details). The heterogeneity test results indicated Q = 2200.62, P < 0.001, I² = 98%, suggesting a high degree of heterogeneity among the included studies. Therefore, a random-effects model was adopted for the evaluation. The effect size serves as an indicator to assess the strength and relevance of the experimental effects. The results of the random-effects model showed an effect size of 0.590 (P < 0.001), indicating a moderate positive effect of SEM on students' physical education learning (detail see Table 3). The details of all analysis results can be found in S5 Appendix: the detailed results of the meta-analysis of the overall and different moderating variables effects.

**Effects of the Sport Education Model on cognitive and non-cognitive dimensions in students.** Table 4 shows that SEM had a small positive effect on students' learning outcomes in the cognitive dimension (Effect size = 0.411, P < 0.042). In the non-cognitive dimension, SEM demonstrated a moderate positive effect on students' learning outcomes (SMD = 0.641, P < 0.01). Both effects reached statistical significance.

**Effects of different moderating variables of the Sport Education Model on student PE learning outcomes.** To explore the impact of different moderating variables on students' physical education learning, we conducted statistical analyses on various Participant characteristics (learning stages, SEM experience, and class sizes) and Intervention characteristics (frequency, duration of each session, and experimental period).

**Effects of the Sport Education Model on student PE learning with different stages.** This study categorized the academic stages of the included studies into elementary, secondary, and university levels. The data in Table 5 indicate that SEM did not achieve statistical significance for physical education learning among elementary school students (P = 0.404), and therefore, SEM cannot be shown to have a significant impact on elementary students' physical education learning. However, SEM demonstrated significant improvements for secondary (Effect size = 1.055; P = 0.003) and

**Table 3. The overall effect of the Sport Education Model on student PE learning.**

| Model | Effect size | Standard error | 95% CI | | Two-tailed test | | Heterogeneity test | | | |
|-------|-------------|----------------|--------|--|-----------------|--|--------------------|--|--|--|
| | | | Lower limit | Upper limit | Z | p | Q | df | p | I² |
| Random model | 0.590 | 0.156 | 0.284 | 0.897 | 3.773 | 0.001* | 2200.62 | 48 | 0.001 | 98% |

**Table 4. Effects of the Sport Education Model on cognitive and non-cognitive dimensions in students.**

| Model | Effect size | Standard error | 95% CI | | Two-tailed test | |
|-------|-------------|----------------|--------|--|-----------------|--|
| | | | Lower limit | Upper limit | Z | p |
| Cognitive dimension | 0.411 | 0.202 | 0.015 | 0.807 | 2.034 | 0.042* |
| Non-cognitive dimension | 0.641 | 0.192 | 0.265 | 1.016 | 3.345 | 0.001* |

**Table 5. Effects of the Sport Education Model on student PE learning with different moderating variables.**

| Coding object | | Effect size | Standard error | 95% CI | | Two-tailed test | |
|---|---|---|---|---|---|---|---|
| | | | | Lower limit | Upper limit | Z | p |
| Stages | elementary | −0.118 | 0.141 | −0.394 | 0.159 | −0.835 | 0.404 |
| | secondary | 1.055 | 0.359 | 0.361 | 1.759 | 2.938 | 0.003** |
| | university | 0.174 | 0.086 | 0.006 | 0.342 | 2.034 | 0.042* |
| Experience | No experience | 0.604 | 0.239 | 0.136 | 1.072 | 2.528 | 0.011* |
| Class sizes | small | 1.058 | 0.379 | 0.314 | 1.802 | 2.788 | 0.005** |
| | medium | 0.791 | 0.306 | 0.192 | 1.390 | 2.586 | 0.010* |
| | large | −0.018 | 0.086 | −0.186 | 0.150 | −0.206 | 0.837 |
| Frequency | ≤1 lessons/week | 0.484 | 0.185 | 0.021 | 0.847 | 2.613 | 0.009** |
| | 2 lessons/week | 1.820 | 0.681 | 0.486 | 3.154 | 2.673 | 0.008** |
| | >2 lessons/week | 0.003 | 0.131 | −0.254 | 0.261 | 0.027 | 0.979 |
| Duration | 30–60 min/session | 0.295 | 0.116 | 0.068 | 0.522 | 2.551 | 0.011* |
| | ≥60 min/session | 1.002 | 0.289 | 0.437 | 1.568 | 3.474 | 0.001** |
| Period | <18 session | 0.654 | 0.182 | 0.297 | 1.010 | 3.591 | 0.001** |
| | ≥18 session | 0.305 | 0.227 | −0.141 | 0.750 | 1.342 | 0.180 |

university (Effect size = 0.174; P = 0.042) students. These results suggest that SEM has a large positive effect on physical education learning among secondary students, with a stronger effect compared to elementary and university students.

**Effects of the Sport Education Model on student PE learning with different experiences.** In the statistical analysis of whether students had SEM experience, only one study reported students with experience. However, meta-analysis was conducted when at least three studies provided sufficient data to calculate the effect size (ES). Additionally, eight studies did not mention whether students had participated in SEM courses, and due to the inability to determine this, these studies were excluded from this part of the analysis. Therefore, we only analyzed students without SEM experience. The data in Table 5 show that SEM had a significant positive effect on the physical education learning of students with no SEM experience (Effect size = 0.604; P = 0.011). This effect size was higher than the overall effect (Effect size = 0.590; P < 0.001).

**Effects of the Sport Education Model on student PE learning with different class sizes.** This section of the study categorized the class sizes in the included studies into small, medium, and large. The data in Table 5 show that SEM did not achieve statistical significance for physical education learning among students in large class sizes (P = 0.837), and therefore, SEM cannot be shown to have a significant impact on students in this class format. However, SEM demonstrated a significant moderate positive effect on students in medium class sizes (Effect size = 0.791; P = 0.010) and a significant large positive effect on students in small class sizes (Effect size = 1.058; P = 0.005). These results suggest that SEM has the greatest positive impact on the physical education learning of students in small class sizes.

**Effects of the Sport Education Model on student PE learning with different intervention frequency.** This section of the study categorized the intervention frequency in the included studies into ≤1 lesson/week, 2 lessons/week, and >2 lessons/week. The data in Table 5 show that SEM did not achieve statistical significance for students with >2 lessons/week intervention (P = 0.979), and therefore, SEM cannot be shown to have a significant impact on students with this intervention frequency. However, SEM with an intervention frequency of ≤1 lesson/week demonstrated a significant small positive effect on students' physical education learning (Effect size = 0.484; P = 0.009), while SEM with an intervention frequency of 2 lessons/week showed a significant large positive effect (Effect size = 1.820; P = 0.008). These results suggest that SEM with an intervention frequency of 2 lessons/week has the greatest positive impact on student's physical education learning.

**Effects of the Sport Education Model on student PE learning with different duration of each session.** This section of the study categorized the duration of each session in the included studies into ≤30 min, 30–60 min/session, and ≥60 min/session. Since no studies used interventions with ≤30 min, the comparison focused on the 30–60 min/session and ≥60 min/session interventions. The data in Table 5 show that SEM interventions with a duration of 30–60 min/session had a significant small positive effect on students' physical education learning (Effect size = 0.295; P = 0.011). However, SEM interventions with a duration of ≥60 min/session demonstrated a significant large positive effect (Effect size = 1.002; P < 0.001). These results suggest that SEM interventions with a duration of ≥60 min/session have the greatest positive impact on students' physical education learning.

**Effects of the Sport Education Model on student PE learning with different experimental periods.** This section of the study categorized the experimental period in the included studies into <18 sessions and ≥18 sessions. The data in Table 5 show that the SEM experimental period of ≥18 sessions did not reach statistical significance for students' physical education learning (P = 0.180), and therefore, it cannot be demonstrated to have a significant impact. However, the SEM experimental period of <18 sessions showed a significant moderate positive effect on students' physical education learning (Effect size = 0.654; P < 0.001). These results suggest that an experimental period of SEM beyond 18 sessions may not be suitable for enhancing students' physical education learning.

## Discussion

Statistical analysis indicates that the Sport Education Model (SEM) can promote students' physical education learning, with positive effects observed in both cognitive and non-cognitive dimensions. Furthermore, the impact of SEM on students' learning outcomes varies across different moderating variables.

The results of this study show that SEM has a significant moderate positive impact on students' physical education learning (Effect size = 0.590, P < 0.001). This finding suggests that SEM, as a structured teaching model, effectively enhances students' cognitive and non-cognitive learning outcomes in physical education courses. This aligns with previous research, such as Manninen & Campbell's (2022) [17] study, which demonstrated SEM's significant role in improving students' basic needs, intrinsic motivation, and prosocial attitudes. Additionally, Bessa et al. (2021) [1] highlighted SEM's positive impact on students' personal and social skills, as well as their motor and cognitive development. The reasons for these effects may include the following aspects: (1) The structural characteristics of SEM and their role in producing positive outcomes [5]. SEM simulates real-life sports event environments and emphasizes teamwork, role-playing, and long-term participation, which are crucial factors contributing to its positive effects [46,47]. Specifically, SEM's structural features typically include season, affiliation, formal competition, culminating event, record keeping, and festivity [1–3,48]. These features not only help develop students' sports skills but also enhance their autonomy and sense of belonging, thus fostering intrinsic motivation [6,20,41]. Additionally, SEM's systematic and coherent approach provides students with ample opportunities for practice, allowing them to consolidate and refine their skills and knowledge across various contexts [2]. (2) The impact of SEM on cognitive and non-cognitive indicators [49,50]. SEM significantly enhances both cognitive and non-cognitive indicators, particularly in improving students' cooperation skills, confidence, and attitudes toward physical education [1,3,43]. This impact may stem from SEM's focus on student-centered autonomy support and social interaction [3]. According to Self-Determination Theory (SDT), students who experience greater autonomy, competence, and relatedness in physical activities tend to exhibit stronger intrinsic motivation and positive emotional experiences [51,52]. Furthermore, SEM emphasizes the cultivation of responsibility and teamwork through competition and role-playing, which is closely linked to the improvement of non-cognitive indicators [9].

Additionally, the impact of SEM on students' physical education learning outcomes varies across different moderating variables. The heterogeneity test results ($I^2 = 98\%$) suggest a high level of variation among the included studies. This high heterogeneity may be attributed to several factors: (1) Differences in participant characteristics, typically including factors

such as age (learning stage), gender, and athletic background [51]; (2) Variations in intervention characteristics, such as differences in the implementation cycle and frequency of SEM, which may lead to varying levels of effect strength [28].

For the purposes of this study, the participant-related moderating variables were categorized into three groups: learning stages, experience, and class sizes. The intervention-related moderating variables were categorized into frequency, duration of each session, and experimental period.

Regarding learning stages, SEM's impact on students' physical education outcomes differed significantly across various stages of learning. Specifically, SEM had a significant large positive effect on secondary school students (Effect size = 1.055, P = 0.003), while its effect on university students was smaller (Effect size = 0.174, P = 0.042), and it did not reach statistical significance for elementary students (P = 0.404). These differences may be closely related to the psychological needs, learning motivation, and cognitive development characteristics of students at different developmental stages [28,53]. The significant effect of SEM on secondary students may primarily stem from SEM's ability to fulfill students' needs for autonomy, competence, and relatedness. According to SDT, secondary students are generally more motivated to achieve a sense of autonomy and team affiliation in their learning [44]. SEM enhances intrinsic motivation and positive emotional experiences significantly through role-playing, team competitions, and continuous contextual teaching [20]. Additionally, secondary students typically possess a certain level of sports skills, enabling them to better understand and execute complex team strategies and role tasks, which further amplifies the positive effects of SEM [48].

Although SEM had a smaller effect on university students (Effect size = 0.174, P = 0.042), the result was still statistically significant. This could be attributed to the strong external motivation for university students' participation in physical education, such as obtaining credits or passing exams [37,45]. While SEM's team collaboration and competition models may stimulate university students' interest, its role-playing and long-duration features may be less appealing to university students who already possess strong self-management skills [37]. Furthermore, university-level physical education courses are typically elective, with limited time, making it difficult to fully implement the extended cycle requirements of SEM [54]. SEM did not significantly impact elementary students (P = 0.404), possibly due to their cognitive development level and the challenges in classroom management [48]. Elementary students are often more reliant on external rewards and teacher guidance, making it difficult for them to understand and perform complex role divisions and team strategies. For example, the role of a referee requires students to have a strong knowledge of rules and practical experience to perform competently.

In terms of learning experience, SEM has a significant positive effect on students with no prior SEM participation experience (Effect size = 0.604, P = 0.011), with its effect size even exceeding the overall effect (Effect size = 0.590, P < 0.001). This result highlights SEM's positive impact on students' physical education learning during their initial exposure, which may be closely related to its novelty and contextual features [36,37]. These characteristics of SEM effectively stimulate students' intrinsic motivation [36], thus enhancing their interest and engagement in learning. According to the Novelty Effect [55,56], students typically exhibit higher levels of interest and involvement when first encountering new teaching methods [57]. Additionally, SEM's contextual and authentic design enables students to better perceive the meaning and value of physical education, thereby fostering sustained learning motivation [3]. Through its seasonal system and role tasks, SEM not only enhances students' sense of participation and confidence [43,44] but also reduces the negative impact of traditional physical education [21], which often overemphasizes physical fitness and competition [3]. As a result, students without prior SEM experience are more likely to accept and integrate this teaching model, leading to significant learning outcomes.

Regarding class sizes, SEM's effects on students' physical education learning outcomes vary significantly across different class sizes. Specifically, SEM has the greatest impact on students in small class sizes (Effect size = 1.058, P = 0.005), showing a significant large positive effect; it also has a significant moderate positive effect on students in medium class sizes (Effect size = 0.791, P = 0.010); however, the effect size for students in large class sizes did not reach statistical significance (P = 0.837). The significant effect of SEM in small class sizes (Effect size = 1.058) can be attributed to several

factors: (1) High-frequency interaction and personalized feedback: Small class sizes allow for more frequent interaction between teachers and students, enabling more targeted feedback [28]. This immediate feedback and high engagement help students gain a stronger sense of competence and achievement in physical education [41]; (2) Role division and contextual authenticity: SEM's emphasis on role-playing (e.g., referee, team captain, coach) is more easily implemented and managed in small classes. Students can better understand and perform role tasks, thereby improving teamwork and self-management skills [28]; (3) Promoting autonomy and relatedness: According to SDT, personalized guidance and role-playing in small classes help satisfy students' needs for autonomy, competence, and relatedness, further enhancing their intrinsic motivation [41,44].

SEM also has a significant moderate effect on students in medium class sizes (Effect size = 0.791, P = 0.010). The advantages of medium class sizes include: (1) Effective resource allocation and teamwork: Medium-sized classes strike a balance between providing students sufficient opportunities for role-playing and practice while effectively utilizing teaching resources (e.g., facilities and equipment) [28]. This resource allocation helps students enhance their physical skills through moderate levels of competition and collaboration [42]; (2) Effective implementation of contextual teaching: SEM's seasonal system and competitive format are easier to implement and manage in medium-sized classes, allowing students to better experience teamwork and contextualized physical learning [43]; (3) Moderate improvement in motivation and engagement: Medium class sizes avoid the pressure associated with the individualized focus of small classes while reducing the issue of "passive participation" common in large classes [7]. Therefore, the lack of significant effects in large class sizes (P = 0.837) may be due to challenges in role distribution and management, insufficient personalized feedback, and the prevalence of "passive participation."

In terms of intervention frequency, SEM demonstrates significant differences in its impact on student's physical education learning outcomes across various intervention frequencies. Specifically, SEM produces a significant large positive effect at an intervention frequency of 2 lessons per week (Effect size = 1.820, P = 0.008), and a significant small positive effect at ≤1 lesson per week (Effect size = 0.484, P = 0.009), whereas no statistically significant effect is observed for frequencies greater than 2 lessons per week (P = 0.979). These findings suggest that an intervention frequency of two lessons per week is optimal for SEM implementation, potentially due to its alignment with students' learning load and adaptability [3,28]. The reasons behind the best effects of 2 lessons per week may include: (1) balanced frequency that ensures recovery and continuity: This frequency allows for adequate recovery while maintaining consistent learning, preventing skill degradation [37]; (2) effectiveness of role-playing and feedback: It provides students multiple opportunities for role-playing and teamwork each week, coupled with timely feedback that enhances interpersonal skills [42], knowledge understanding [20], game performance [37], physical activity (PA) [20], autonomy [41,44], and competence [41,44]; (3) avoidance of overtraining effects: Higher frequencies (>2 lessons per week) may lead to fatigue and burnout, ultimately diminishing learning outcomes [3]. The lower effects observed at ≤1 lesson per week can be attributed to: (1) extended learning intervals that hinder skill retention: The infrequent sessions make it challenging for students to maintain continuous learning, leading to decreased skill retention and team synergy [20,40]; (2) insufficient role-playing frequency: The low frequency of role-playing and contextual teaching struggles to foster students' identified regulation and external regulation, thereby undermining learning effectiveness [20,40].

In terms of session duration, SEM produces a significant large positive effect with intervention sessions lasting ≥60 minutes (Effect size = 1.002, P < 0.001), whereas a significant small positive effect is observed with session durations of 30–60 minutes (Effect size = 0.295, P = 0.011). These results suggest that sessions exceeding 60 minutes are most effective for SEM, potentially due to the time demands of role-playing and the complexity of teamwork. The reasons behind the optimal effect at ≥60 minutes per session include: (1) sufficient time for role-playing and feedback: The role-playing and competitive elements emphasized by SEM require extended periods for students to fully experience role tasks and receive feedback [37,38]; (2) necessity of teamwork and strategy execution: Longer sessions provide adequate time for students to implement team strategies, improving collaboration and responsibility [5,37]; (3) alignment with Cognitive

Load Theory [58]: Extending the session duration allows students to process complex tactics and rules gradually, preventing overload [36]. The lower effects observed in the 30–60 minute sessions may be attributed to: (1) insufficient time for role task execution: Shorter sessions limit the depth of role-playing and teamwork, hindering students from fully experiencing the contextual nature of the physical education curriculum [42]; (2) inadequate feedback mechanisms: The brevity of the sessions makes it difficult for teachers to provide timely, targeted feedback, leading to less effective skill consolidation [8].

In terms of intervention period, SEM produces a significant moderate positive effect with intervention periods of no more than 18 sessions (Effect size = 0.654, P < 0.001), whereas no statistically significant effect is observed for periods of ≥18 sessions (P = 0.180). Although these results are inconsistent with recommended intervention periods from previous studies [3,5], the present study argues that a period of no more than 18 sessions is a reasonable duration for SEM implementation, which may be closely related to students' motivation sustainability and the complexity of the teaching content [35,37,39]. The reasons for the optimal effects at ≤18 sessions may include: (1) novelty of short-term interventions and enhanced motivation: Shorter periods help maintain students' interest in role-playing and competition, avoiding fatigue and burnout associated with prolonged intervention periods [38,44]; (2) clear task objectives and timely feedback: Shorter interventions enable students to receive feedback more quickly, clarify goals and tasks, thereby enhancing intrinsic motivation [20]; (3) avoidance of the burnout effect: Longer intervention periods may lead to students losing interest in their roles and tasks, which undermines learning outcomes [28]. The lack of significance for periods ≥18 sessions may be attributed to: (1) insufficient motivation sustainability: Extended periods may result in repetitive role-playing and teamwork experiences, leading to motivation decline [21,37,39]; (2) excessive cognitive load, which affects learning outcomes: The complexity of tactics and role tasks in long-term interventions may overwhelm students' cognitive load, thus negatively impacting learning outcomes [59].

### Limitations

Several limitations in this study warrant attention. First, due to the limited number of studies on each moderator variable, subgroup analyses of cognitive and non-cognitive indicators within each moderator variable were not conducted. Furthermore, the small number of studies included in the meta-analysis (n = 6) that assessed the impact of SEM on non-cognitive indicators may lead to potential discrepancies between the meta-analysis results and the actual outcomes. Second, no studies have examined potential confounding factors that may influence students' physical education learning, such as personality, emotions, sleep behaviors, and dietary habits. Third, according to the GRADE assessment, the certainty of the reported cognitive and non-cognitive outcomes is very low, which undermines the credibility of these estimates. Despite these limitations, our study provides novel and valuable practical insights for frontline physical education teachers and researchers, revealing the positive impact of SEM on enhancing students' physical education learning.

### Practical applications

The results of this study reveal significant differences in the effects of SEM on students' physical education learning outcomes under various teaching conditions, providing several practical implications. First, with regard to participant characteristics, it is recommended that frontline physical education teachers implement SEM interventions with small class sizes for secondary students who lack prior SEM experience, as this approach demonstrates the best outcomes. Regarding intervention characteristics, it is suggested that frontline physical education teachers design SEM course plans that involve two lessons per week, with each session lasting no less than 60 minutes and a total of no more than 18 lessons.

### Conclusion

In conclusion, SEM can effectively promote students' physical education learning and positively impact their cognitive and non-cognitive abilities. Specifically, the most effective SEM intervention plan involves two lessons per week, each lasting

no less than 60 minutes and no more than 18 lessons, applied to secondary students with no prior SEM experience in small class sizes. This study provides frontline physical education teachers with recommendations for selecting participant and intervention characteristics for SEM implementation and offers theoretical support for the high-quality development of school sports through SEM.

## Supporting information

**S1 Appendix. PRISMA 2020 checklist.**
(PDF)

**S2 Appendix. Detailed search strategy.**
(DOCX)

**S3 Appendix. Description of 430 records.**
(XLS)

**S4 Appendix. Data used for meta-analysis.**
(XLSX)

**S5 Appendix. The detailed results of the meta-analysis of the overall and different moderating variables effects.**
(DOCX)

## Author contributions

**Data curation:** Gege Yao.

**Funding acquisition:** Wensheng Xiao.

**Methodology:** Xiaorong Bai, Wensheng Xiao.

**Resources:** Wensheng Xiao.

**Software:** Xiaorong Bai.

**Supervision:** Kim Geok Soh, Mohd Ashraff Mohd Anuar.

**Writing – original draft:** Junlong Zhang.

**Writing – review & editing:** Kim Geok Soh, Lixia Bao.

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
