## [Decision Letter · Decision Letter 0]

24 Jun 2025

Dear Dr. zhang,

Thank you for submitting your manuscript to PLOS ONE. After careful consideration, we feel that it has merit but does not fully meet PLOS ONE’s publication criteria as it currently stands. Therefore, we invite you to submit a revised version of the manuscript that addresses the points raised during the review process.

We look forward to receiving your revised manuscript.

Kind regards,

Mert Kurnaz

Guest Editor

PLOS ONE

Journal Requirements:

[This study was supported by A Project Supported by the Education Ministry's Youth Fund Project for Humanities and Social Sciences Research of China (Grant No. 24YJC890001).].

4. Thank you for stating the following in your manuscript:

[This study was supported by A Project Supported by the Education Ministry's Youth Fund Project for Humanities and Social Sciences Research of China (Grant No. 24YJC890001).]

[This study was supported by A Project Supported by the Education Ministry's Youth Fund Project for Humanities and Social Sciences Research of China (Grant No. 24YJC890001).]

5. Please include captions for your Supporting Information files at the end of your manuscript, and update any in-text citations to match accordingly. Please see our Supporting Information guidelines for more information: http://journals.plos.org/plosone/s/supporting-information .

Additional Editor Comments:

The manuscript presents a meta-analysis investigating the impact of the Sport Education Model (SEM) on students’ physical education learning outcomes, with a focus on participant and intervention characteristics. The topic is highly relevant, especially for educators and policymakers seeking evidence-based practices in physical education. The study is well-structured, follows PRISMA guidelines, and attempts to contribute to understanding moderating factors that affect SEM efficacy.

However, while the paper is ambitious and has notable strengths, several critical concerns related to methodological rigor, reporting transparency, analytical limitations, and writing quality must be addressed.

You can see my suggestions to authors below:

Certainty of Evidence (GRADE Assessment) – Serious Concern

The GRADE evaluation rates the certainty of evidence for both cognitive and non-cognitive outcomes as very low. This severely limits the confidence in the findings.

The authors should either:

a) justify why their conclusions are still meaningful despite low evidence certainty, or

b) temper their recommendations and claims of practical applicability.

The I² value of 98% indicates extremely high heterogeneity, which threatens the validity of the overall pooled estimate.

Although subgroup analyses are presented, the authors must explore and report sources of heterogeneity more deeply (e.g., study design quality, cultural context, variation in implementation fidelity).

More visual representation of heterogeneity in the main manuscript or supplementary materials would strengthen transparency.

The subgroup analyses (e.g., by session frequency or class size) are informative but risk Type I error due to multiple comparisons and small subgroup sizes.

The authors should indicate whether interaction tests were performed between subgroups and report corresponding p-values.

The classification of outcomes into “cognitive” and “non-cognitive” needs clearer operational definitions.

It is currently unclear, for instance, why “motor skills” are categorized under cognitive outcomes when they also depend on psychomotor learning.

While RoB-2 and ROBINS-I were used, the specific domains of bias (e.g., blinding, selection bias) are not discussed in sufficient depth.

Figure references are made (e.g., Fig 2, Fig 3), but visuals are not included in the provided material. Ensure these are present in the final submission.

The inclusion of non-English language articles is commendable, but more detail is needed about how translations were verified and whether this affected coding reliability.

While overall readable, the manuscript contains several grammatical errors and awkward phrasings.

The term “learning efficiency” is used without clear definition and appears non-standard. Consider replacing with “learning outcomes” or “educational gains.”

Some appendices (e.g., Appendix 3 – description of 430 records) are critical for replication but should be summarized in the main text or tables for clarity.

The search was concluded in February 2025, and some included studies are listed as from 2024. Confirm the accuracy and peer-reviewed status of these articles.

The authors recommend 2 lessons/week for ≤18 sessions universally. However, contextual constraints (e.g., curriculum length, teacher capacity) should be acknowledged.

Recommendation: Minor Revision

This study presents an important contribution to physical education literature by quantifying SEM’s impact across various implementation conditions. However, to ensure robustness and clarity, substantial revisions are required, particularly regarding methodological transparency, heterogeneity management, and evidence strength communication.

Reviewers' comments:

Reviewer's Responses to Questions

**Comments to the Author**

1. Is the manuscript technically sound, and do the data support the conclusions?

Reviewer #1: Partly

Reviewer #2: Yes

2. Has the statistical analysis been performed appropriately and rigorously?

Reviewer #1: N/A

Reviewer #2: Yes

3. Have the authors made all data underlying the findings in their manuscript fully available?

Reviewer #1: Yes

Reviewer #2: Yes

4. Is the manuscript presented in an intelligible fashion and written in standard English?

Reviewer #1: Yes

Reviewer #2: Yes

Reviewer #1: The manuscript presents relevant findings on the effectiveness of the Sport Education Model. The methodology is generally appropriate, and the analysis is clear. Please ensure that all journal formatting and reference guidelines are carefully followed before final submission.

Reviewer #2: I had the pleasure of participating in the review process of a manuscript entitled "Effective Implementation of the Sport Education Model in Physical Education: A Meta-Analysis of Participant and Intervention Characteristics."

The study investigates the facilitative effects of SEM on students' physical education learning and examines the Participant and Intervention Characteristics that modulate its impact.

The study provides novel research findings that significantly enhance the existing knowledge on the subject. A series of meticulously designed experiments were conducted, and the resulting data underwent rigorous statistical analyses, all executed to elevated technical standards. The author provides sufficient detail about the methods used in these analyses to ensure their comprehension and replication. The study's conclusions are obvious and based on the evidence. The author writes the document in typical academic English, demonstrating careful attention to word choice and order. It also follows the correct regulations for reporting (CONSORT, PRISMA...).

For these reasons, I believe that the manuscript is suitable for publication in its current form.

**Do you want your identity to be public for this peer review?** For information about this choice, including consent withdrawal, please see our Privacy Policy

Reviewer #1: **Yes: ** Dr. Swamynathan Sanjaykumar

Reviewer #2: No

---

## [Author Response · Author response to Decision Letter 1]

7 Aug 2025

Thank you for your detailed feedback and guidance. We have:

We have carefully reviewed the latest version of my manuscript and confirmed that there is no funding-related text in the main text, acknowledgments, or other sections. The only funding information provided is the Funding Statement in the online submission form, which currently reads:

This study was supported by the Education Ministry's Youth Fund Project for Humanities and Social Sciences Research of China (Grant No. 24YJC890001).

We appreciate your assistance and have addressed all points carefully in our resubmission.

We sincerely thank the editorial team and reviewers for their thoughtful and constructive feedback. Although we were granted sufficient time for revision, we devoted extensive effort to revising the manuscript promptly and thoroughly. All suggested modifications have been carefully addressed, and the manuscript has undergone two rounds of internal review to ensure accuracy and completeness.

Journal Requirements:

1.Please ensure that your manuscript meets PLOS ONE's style requirements, including those for file naming.

Response:

Thank you for the comment. We have carefully revised the manuscript to ensure full compliance with PLOS ONE’s formatting requirements. Specifically, all headings have been adjusted to use sentence case and appropriate font size, and all file names have been updated according to the journal’s naming guidelines. These changes are visible in the formatting of the headings throughout the manuscript. Additionally, we reformatted the figures following the FITT format; figures have been removed from the main text (only titles remain), and all figures have been uploaded as separate files. Figure titles are now placed immediately after the paragraph where each figure is first cited.

2–4 (Funding Statement):

3. Thank you for stating in your Funding Statement: [This study was supported by A Project Supported by the Education Ministry's Youth Fund Project for Humanities and Social Sciences Research of China (Grant No. 24YJC890001).] Please provide an amended statement that declares all the funding or sources of support (whether external or internal to your organization) received during this study, as detailed online in our guide for authors. Please also include the statement “There was no additional external funding received for this study.” in your updated Funding Statement.

4. We note that you have provided funding information that is currently declared in your Funding Statement. However, funding information should not appear in the Acknowledgments section or other areas of your manuscript. Please remove any funding-related text from the manuscript and let us know how you would like to update your Funding Statement.

Response:

Thank you for your detailed comments regarding the funding information. We have carefully addressed each of the issues raised as follows:

We have corrected the inconsistency between the ‘Funding Information’ and the ‘Financial Disclosure’ sections. The grant number [Grant No. 24YJC890001] now appears consistently and correctly in the ‘Funding Information’ section.

We have revised the Funding Statement to declare all sources of support received during the study. The updated Funding Statement is as follows:

Funding:

This study was supported by the Education Ministry's Youth Fund Project for Humanities and Social Sciences Research of China (Grant No. 24YJC890001). There was no additional external funding received for this study.

In accordance with journal policy, we have removed the funding information from the Acknowledgments section of the manuscript. Funding is now declared exclusively in the Funding Statement section, as requested.

We have included the revised funding statement in the cover letter. Please update the online submission form on our behalf.

We hope these revisions meet your requirements and sincerely appreciate your guidance in helping us align with PLOS ONE’s submission standards.

5.Please include captions for your Supporting Information files at the end of your manuscript, and update any in-text citations to match accordingly. Please see our Supporting Information guidelines for more information.

Response:

Thank you for your comment. We have now added clear and complete captions for all Supporting Information files at the end of the manuscript, as per the journal’s guidelines. In addition, all in-text citations referring to Supporting Information have been updated to ensure consistency and correct referencing.

6.“Please review your reference list to ensure that it is complete and correct. If you have cited papers that have been retracted, please include the rationale for doing so in the manuscript text, or remove these references and replace them with relevant current references…”

Response:

Thank you for your comment. We have carefully reviewed the entire reference list to ensure accuracy and completeness. No retracted articles are currently cited. Minor formatting corrections have been made where necessary, and a few outdated references have been replaced with more recent and relevant sources (e.g. No. 19). These changes have been reflected in both the revised manuscript and the updated reference list.

Additional Editor Comments:

1.The manuscript presents a meta-analysis investigating the impact of the Sport Education Model (SEM) on students’ physical education learning outcomes, with a focus on participant and intervention characteristics. The topic is highly relevant, especially for educators and policymakers seeking evidence-based practices in physical education. The study is well-structured, follows PRISMA guidelines, and attempts to contribute to understanding moderating factors that affect SEM efficacy.

However, while the paper is ambitious and has notable strengths, several critical concerns related to methodological rigor, reporting transparency, analytical limitations, and writing quality must be addressed.

Response:

Dear editor, on behalf of my co-authors, I would like to sincerely thank you for your thoughtful and constructive comments regarding our manuscript. We are grateful that you found the topic of our meta-analysis relevant and timely, especially for educators and policymakers who rely on evidence-based insights in physical education. Your acknowledgment of the manuscript's structure, adherence to PRISMA guidelines, and its attempt to investigate moderating factors influencing the efficacy of the Sport Education Model (SEM) is deeply appreciated.

At the same time, we take your critical feedback regarding methodological rigor, reporting transparency, analytical limitations, and writing quality with utmost seriousness. In response to these concerns, we have undertaken a comprehensive revision of the manuscript. Below are our detailed point-by-point responses to the revisions you suggested.

2.Certainty of Evidence (GRADE Assessment) – Serious Concern

“The GRADE evaluation rates the certainty of evidence for both cognitive and non-cognitive outcomes as very low. This severely limits the confidence in the findings. The authors should either: a) justify why their conclusions are still meaningful despite low evidence certainty, or b) temper their recommendations and claims of practical applicability.”

Response:

We appreciate the editor’s critical insight regarding the very low certainty of evidence as indicated by our GRADE assessment. In response, we have taken two important steps:

(1)Justification of Significance Despite Low Certainty:

While the GRADE assessment revealed low confidence in the estimates due to issues such as risk of bias and heterogeneity, the statistically significant and consistent effect sizes observed across both cognitive and non-cognitive domains suggest meaningful practical relevance. These findings provide valuable directional evidence for educators and policymakers, particularly in the context of limited high-quality randomized trials in this area.

(2) Tempering Practical Claims:

We have revised several sections of the manuscript, especially the Abstract, Discussion, and Conclusion, to soften the strength of our recommendations. Phrases implying definitive or causal conclusions have been moderated, and we have clearly acknowledged the limitations posed by the low GRADE ratings. Additionally, we now emphasize the exploratory nature of our meta-analysis and recommend further high-quality studies to strengthen future evidence.

The revised parts in the manuscript:

Abstract (Page 3):

Original: “This study offers practical recommendations for SEM implementation and theoretical support for the high-quality development of school sports.”

Revised: “This study offers preliminary recommendations for SEM implementation and provides exploratory insights that may inform the high-quality development of school sports, while acknowledging the current limitations in evidence certainty.”

Limitations (Page 33)

Original: “...revealing the positive impact of SEM on enhancing students' physical education learning.”

Revised: “...revealing the potential positive impact of SEM on enhancing students' physical education learning, although these findings should be interpreted with caution due to the low certainty of evidence.”

Conclusion (page 35):

Original: “This study provides frontline physical education teachers with recommendations for selecting participant and intervention characteristics for SEM implementation and offers theoretical support for the high-quality development of school sports through SEM.”

Revised: “This study provides preliminary guidance for frontline physical education teachers regarding SEM implementation under different conditions. However, due to the very low certainty of evidence, these suggestions should be interpreted cautiously and require further validation through high-quality randomized trials.”

3.High Heterogeneity (I² = 98%) and Need for Deeper Exploration

“The I² value of 98% indicates extremely high heterogeneity, which threatens the validity of the overall pooled estimate. Although subgroup analyses are presented, the authors must explore and report sources of heterogeneity more deeply (e.g., study design quality, cultural context, variation in implementation fidelity).

Response:

We fully acknowledge the extremely high heterogeneity (I² = 98%) and agree that this presents a challenge to the validity of pooled estimates. In response to your suggestion, we have taken the following actions:

We expanded our Methods, Results and Discussion sections to include deeper exploration of potential sources of heterogeneity. Specifically, we added subgroup analyses based on study design and variation in implementation fidelity (Methods: Page 11; Results: Page17 and Page 21-24; Discussion: Page 32-33).

4.More visual representation of heterogeneity in the main manuscript or supplementary materials would strengthen transparency.

Response:

Thank you for your helpful suggestion. In response, we have included additional visual representations of heterogeneity in the supplementary material. Specifically, S3 File (“The detailed results of the meta-analysis of the overall and different moderating variables effects”) now contains forest plots of subgroup comparison to enhance the transparency and interpretability of heterogeneity across studies.

5.The subgroup analyses (e.g., by session frequency or class size) are informative but risk Type I error due to multiple comparisons and small subgroup sizes.

Response:

Thank you for pointing out this important concern. In response, we have revised both the Results and Discussion sections to explicitly acknowledge the increased risk of Type I error associated with multiple subgroup comparisons and small sample sizes within subgroups.

In the Results section, we added the following statement:“Nevertheless, due to the small number of studies within the intervention frequency and class size subgroups, there is an increased risk of Type I error, and thus these findings should be interpreted with appropriate caution.”(Page 20)

In the Discussion section, we further addressed this issue by noting:

“The extremely high heterogeneity (I² = 98%) suggests that differences in study design and implementation fidelity may have significantly influenced the pooled estimates. Although subgroup analyses were conducted, the exploratory nature of these analyses must be acknowledged—particularly given the elevated risk of Type I error. Future meta-analyses should aim to more precisely model these moderators.” (page 32-33: Line 620-630)

We believe these revisions strengthen the transparency and methodological caution of our interpretations and align with best practices in meta-analytic reporting.

6.The authors should indicate whether interaction tests were performed between subgroups and report corresponding p-values.

Response:

We sincerely appreciate this insightful suggestion. Conducting interaction tests between subgroups would indeed provide a more rigorous understanding of whether the observed subgroup differences are statistically meaningful. However, due to the limited number of studies within certain subgroups and the resulting low statistical power, we chose not to perform formal interaction tests in the current analysis to avoid overinterpretation of unstable estimates.

That said, we fully acknowledge the value of such analyses and agree that future meta-analyses with a larger pool of eligible studies would benefit from including formal interaction tests to better identify and quantify potential effect modifiers. We have now mentioned this as a direction for future research in the Discussion section.

7.The classification of outcomes into “cognitive” and “non-cognitive” needs clearer operational definitions. It is currently unclear, for instance, why “motor skills” are categorized under cognitive outcomes when they also depend on psychomotor learning.

Response:

We sincerely appreciate the reviewer’s thoughtful comment regarding the clarity of our outcome classification. We acknowledge that, to date, the academic community has not yet reached a universally agreed-upon distinction between “cognitive” and “non-cognitive” outcomes, particularly in the context of physical education research.

To address this concern, we have provided explicit operational definitions of both cognitive and non-cognitive outcomes under the “Eigenvalue coding and data extraction” subsection (page 10, highlighted in red in the revised manuscript). Our classification approach is consistent with that used in previous meta-analyses, particularly the work by Zhao et al. (2024), who categorized similar outcome variables along cognitive and non-cognitive lines within the context of physical education. This study has been cited as reference [28] in our manuscript:

Zhao, M., Lu, X., Zhang, Q., et al. (2024). Effects of exergames on student physical education learning in the context of the artificial intelligence era: a meta-analysis. Scientific Reports, 14(1), 7115. https://doi.org/10.1038/s41598-024-57357-8

We believe this clarification improves the transparency of our coding strategy and aligns our work with prior literature in this field.

8.While RoB-2 and ROBINS-I were used, the specific domains of bias (e.g., blinding, selection bias) are not discussed in sufficient depth.

Response:

Thank you for this insightful observation. We agree that providing a deeper discussion of the specific domains assessed by RoB-2 and ROBINS-I would strengthen the rigor and interpretability of our findings. In response, we have added two dedicated paragraphs (Page 23-24: Line 417-434) in the Discussion section to reflect on how common sources of bias-such as random sequence generation, lack of blinding, confounding, and missing data-may have influenced the effect estimates in the included studies.

This discussion elaborates on the differential risk profiles observed across randomized an

---

## [Editor Report · Decision Letter 1]

13 Aug 2025

Effective Implementation of the Sport Education Model in Physical Education: A Meta-Analysis of Participant and Intervention Characteristics

PONE-D-25-13280R1

Dear Dr. Zhang,

We’re pleased to inform you that your manuscript has been judged scientifically suitable for publication and will be formally accepted for publication once it meets all outstanding technical requirements.

Kind regards,

Mert Kurnaz, Ph.D

Guest Editor

PLOS ONE

Additional Editor Comments (optional):

I have carefully reviewed your responses to the reviewers’ comments and the changes you have made. The revisions have addressed all the points raised in the review process with clarity and thoroughness. The manuscript is now stronger, both in methodological transparency and in the depth of its discussion, and it clearly communicates its contribution to understanding how participant and intervention characteristics influence the Sport Education Model’s effectiveness. Thank you for your diligence and professionalism throughout the revision process, and congratulations on producing a valuable and well-crafted contribution to the literature.

---

## [Editor Report · Acceptance letter]

PONE-D-25-13280R1

PLOS ONE

Dear Dr. zhang,

I'm pleased to inform you that your manuscript has been deemed suitable for publication in PLOS ONE. Congratulations! Your manuscript is now being handed over to our production team.

Kind regards,

on behalf of

Dr. Mert Kurnaz

Guest Editor

PLOS ONE